# Identification of Potential Habitats and Adjustment of Protected Area Boundaries for Large Wild Herbivores in the Yellow-River-Source National Park, China

**Shengwang Bao and Fan Yang ***

School of Economic and Management, Zhejiang Ocean University, Zhoushan 316022, China; baoshengwang@zjou.edu.cn
* Correspondence: yang-fan@zjou.edu.cn

**Abstract:** The wild large herbivores inhabiting the Yellow-River-Source National Park (YRSNP) are confronted with a significant threat from climate change and human activities. In response to these detrimental influences, measures have been proposed by the government, such as the Ecological Conservation and Restoration Project in the Sanjiangyuan Region (ECRPSR) and the establishment of the Sanjiangyuan National Park (SNP). To advance species diversity, it is crucial to investigate the spatial distribution of large herbivores, identify factors influencing their distribution, and address conflicts arising from divergent plans within the YRSNP. In this study, unmanned aerial vehicles were employed for surveying the distribution of the Tibetan wild ass *(Equus kiang)* and Tibetan gazelle (*Procapra picticaudata*). The findings indicate that the optimal habitat area for Tibetan wild ass is 437.16 km$^2$, while for Tibetan gazelle, it is 776.46 km$^2$. Precipitation and the human footprint index emerge as the primary factors influencing the habitat distribution of large herbivores within the YRSNP. Under the influence of the ECRPSR, there was a noteworthy expansion of the habitat area for Tibetan wild ass by 791.25 km$^2$, and for Tibetan gazelle, it expanded by 1612.94 km$^2$. From a wildlife conservation standpoint, this study proposes the establishment of a wildlife refuge in the YRSNP, effective coordination of conflicts between various functional zones and plans, preservation of suitable habitats for large herbivores, and the provision of a scientific foundation to reconcile development and conservation conflicts in the region, while concurrently fostering biodiversity conservation.

**Keywords:** species distribution modeling; suitable habitat; yellow-river-source national park; nature reserve planning

## 1. Introduction

Climate change is a pressing global environmental issue that poses a significant threat to biodiversity worldwide [1,2]. Its impacts extend to the geographical distribution and population sizes of wildlife [3], as it shifts habitats [4], living conditions [5], and species interactions [6]. In addition, climate change disrupts the distribution patterns of numerous species [7,8] and hampers the connectivity between their habitats, resulting in the disappearance of ecological corridors and impeding species migration [9], potentially leading to species extinctions [10,11]. Research indicates that a global temperature increase of 2 °C could trigger the extinction of 15–35% of species [12]. Therefore, addressing global climate change and devising effective strategies to protect species diversity have become paramount concerns for the general public, governments, and the scientific community [13].

With the progress of the economy and society, the incessant expansion of human endeavors not only disrupts the ecosystem [14], but also engenders formidable perils, such as the diminishment of wild animal habitats [10,15] and interspecific species competition [16]. Instances where human activities overlap with those of wild animals often result in conflict [17]. The construction of roads and the expansion of land use manifest as direct ramifications of human activities [18], exerting deleterious influences that primarily





contribute to habitat fragmentation and species extinction [19], thereby exerting a profound impact on biodiversity. Therefore, human-induced disturbance emerges as a salient factor necessitating consideration during the process of species distribution modeling (SDM).

The Sanjiangyuan region, situated in the hinterland of the Qinghai–Tibet Plateau stands as the plateau's most biodiverse region [20]. Over the past few decades, global climate change has significantly impacted the species within the Sanjiangyuan region [21]. As a region of utmost sensitivity and critical significance in terms of climate change on a regional, hemispheric, and even global scale, the Sanjiangyuan region endures recurring ecosystem deterioration and extensive human intervention [22]. Research findings have emphasized the importance of establishing protected areas as a conservation strategy to address the challenges posed by climate change and human activities, which impinge upon species diversity [23,24]. Henceforth, in 2005, the State Council of China initiated the Ecological Conservation and Restoration Project in the Sanjiangyuan Region (ECRPSR), which entered its second phase in 2014, aiming to establish nature reserves to restore natural ecosystems and bolster species preservation. Additionally, in 2014, the State Council of China unveiled its inaugural national park plan, the Sanjiangyuan National Park (SNP), with the objective of erecting a national park to protect biodiversity and enhance ecological system services. Nevertheless, challenges persist regarding functional zoning, encompassing issues such as ambiguous definitions of rights and responsibilities, boundary ambiguity, and conflicts, alongside the exclusion of significant habitats for wildlife within the nature reserves [23]. Therefore, it becomes urgent, from a wildlife conservation standpoint, to address and reconcile the conflicts that arise between nature reserves and functional zones.

The Yellow-River-Source National Park (YPSNP) emerges as one of the protected areas outlined in the SNP. Featuring a unique plateau and alpine climate, it encompasses an exceptional alpine natural ecosystem, exerting a pivotal role in global biodiversity preservation [25]. Recent studies have demonstrated that the large wild herbivores in the Sanjiangyuan region have experienced significant impacts due to global climate fluctuations [26]. The primary large wild herbivores within the YRSNP are the Tibetan gazelle *(Procapra picticaudata)* and the Tibetan wild ass *(Equus kiang)*. Presently, the Tibetan gazelle holds the designation of being a "Near Threatened species" according to the IUCN institution, which is classified as a national second-class protected animal in China. The Tibetan wild ass stands as the sole wild ungulate species on the plateau and is accorded the status of a national first-class protected animal by China. Moreover, the expansion of pastures, road construction, and the implementation of fences, stemming from human activities, have further intensified the threats faced by the populations of these two species. These threats encompass conflicts between wildlife and livestock, habitat loss, and hindrances to migration routes [27]. Therefore, conducting surveys and monitoring the population dynamics and habitat distribution patterns of large wild herbivores in the YRSNP proves instrumental in implementing scientifically grounded measures to protect species diversity.

The deployment of unmanned aerial vehicles (UAVs) for the investigation of wild animal geographical distribution represents a novel approach to species occurrence surveys. In comparison to the conventional surface line transect survey method, UAV employment addresses its inherent limitations, which encompass low efficacy, exorbitant costs, mutual obstruction of survey targets, constrained survey routes dictated by ground conditions, and the challenge of corroborating survey results consistently. Therefore, it facilitates the fulfillment of requirements pertaining to the long-term monitoring of wild animal populations [28]. Owing to its advantageous features, such as precise enumeration, minimal disturbance to wildlife, and terrain independence, the UAV methodology has witnessed extensive application in the survey of wildlife species distribution, specifically in identifying specific geographical locations [29]. In order to update the distribution areas of prominent large herbivores within the YRSNP, 14 survey sample routes were established utilizing UAVs to investigate the Tibetan wild ass and Tibetan gazelle populations.

Studying the impacts of climate change and human activities on the habitats of endangered species, particularly the large wild herbivores in the YRSNP, holds significant importance for effectively preserving biodiversity and ecosystem integrity. Assessing suitable habitats for target species serves as a crucial initial step in implementing conservation measures [30]. The MaxEnt model conceptualizes species and their habitat environment as a dynamic system characterized by energy dissipation, wherein such changes in dissipation contribute to increased entropy [31]. By calculating the state parameters at which the system reaches maximum entropy, it is possible to determine the relatively stable relationship between species and their environment. Due to its simplicity and minimal sample requirements [32], the MaxEnt model produces superior prediction results. Notably, the algorithm incorporates all available data while avoiding assumptions regarding unknown data, thereby ensuring objectivity and accuracy in predictions [33]. Currently, the MaxEnt model finds widespread application in habitat assessments of endangered species, such as the pangolin [34], Bengal tiger [35], and ungulate [26], etc.

This research employed UAVs to investigate the geographical occurrence sites of large wild herbivores in the YRSNP. The MaxEnt model was then utilized to analyze changes in the habitat distribution and ecological corridors of the Tibetan wild ass and Tibetan gazelle. This analysis considered environmental factors such as climate change and human activities. The primary objectives of this research are as follows: (1) To determine the species distribution pattern of the Tibetan wild ass and Tibetan gazelle in the YRSNP. (2) To identify the main factors influencing the habitat of these two species. (3) To propose adjustments to the functional zoning area based on the distribution of species abundance.

## 2. Materials and Methods

### 2.1. Study Area and Investigated Species

The YRSNP, situated within the SNP in China, spans from 97°1′20″ to 99°14′57″ E and 33°55′5″ to 35°28′15″ N. It covers an area of 19,100 km², including Yellow River Township, Zhaling Lake Township, Machar Town, and 19 administrative villages in Maduo County. The park comprises three distinct regions: the core conservation area (80,600 km²), the ecological conservation and restoration area (40,400 km²), and the traditional utilization area (80,100 km²).

The park (Figure 1) is characterized by an abundance of rivers and lakes, notably Zhaling Lake and Eling Lake, which are the largest natural lakes in the upper reaches of the Yellow River. Along with the Xingxinghai area and other lake clusters, they collectively form the iconic "thousand lakes" landscape of the Yellow River source. Situated at an altitude of 4164 to 4414 m, the region experiences a typical plateau continental alpine climate, characterized by an average annual temperature ranging from −7.03 to −1.13 °C, an annual precipitation between 3060.6 and 5485.8 mm, and distinct hot and cold seasons with well-defined dry and wet periods. Abundant sunshine and strong radiation are prevalent in the area. Alpine vegetation, including meadows, grasslands, shrubs, and swamps, thrives across the region. The unique ecosystem of alpine wetlands and grasslands primarily encompasses the vast plateau lake wetlands of Zhaling Lake, Eling Lake, and the Xingxinghai area.

Within the park, various wild animals find refuge, such as the Tibetan wild ass *(Equus kiang)*, Tibetan gazelle *(Procapra picticaudata)*, brown bear *(Ursus arctos)*, snow leopard *(Panthera uncia)*, wolf *(Canis lupus)*, black-necked crane *(Grus nigricollis)*, eagle *(Accipitridae)*, red duck *(Tadorna ferruginea)*, and bar-headed goose *(Anser indicus)*. The Tibetan wild ass is a large wild herbivore on the Qinghai–Tibet Plateau. Its body shape is similar to that of the Mongolian kulan *(Equus hemionus Pallas)*, with deep ear tips, dorsal ridges, a mane, and tail tips. It feeds on white grass, carex, and various stipa [36]. The Tibetan gazelle is a typical alpine species and cold desert animal, living in alpine meadows, subalpine steppe meadows, and alpine desert areas between 300 and 5750 m above sea level. It feeds mainly on sedge and gramineous plants, artemisia, and other grasses. The Tibetan gazelle

is 91–105 cm in length, with a short and wide snout, a protrusion forehead, large and round eyes, short ears, and a short tail [37].

**Figure 1.** Study Area and the location of investigated species in Yellow-River-Source National Park.

*2.2. The Selection of Species Occurrence Point Data*

Between 2017 and 2021, 4 UAVs including 1 electric fixed-wing UAV, 1 Feima F1000 UAV, and 2 Fuel-powered fixed-wing UAVs were utilized to extensively survey wildlife geographic distribution and population in Maduo County, located in the Sanjiangyuan region of the Qinghai–Tibet Plateau, Qinghai Province, China. The parameters of the UAVs used can be seen in the Table 1. In a field survey, the method of systematic sampling was determined, and 14 UAV sample routes were identified, as shown in Figure 1. The total route length was 120.93 km$^2$, and the effective shooting area was 356 km$^2$. We recorded the species, quantity, and locations of wild animals, encompassing 379 geographic coordinates of Tibetan wild ass and 199 geographic coordinates of Tibetan gazelle. In addition, 3 and 6 geographic coordinates for the Tibetan wild ass and Tibetan gazelle, respectively, were added to the Global Biodiversity Information Database (GBIF, https://www.gbif.org/, accessed on 23 July 2023). To mitigate spatial autocorrelation from dense point sites, points within a 100 m distance were excluded. The remaining points were modeled to represent species distribution. The geographic coordinates finally comprised 79 Tibetan wild ass and 52 Tibetan gazelle.

**Table 1.** The parameters of UAVs.

| Parameters | Electric Fixed-Wing UAV | Fuel-Powered Fixed-Wing UAV | Feima F1000 UAV |
|---|---|---|---|
| Number of aircrafts | 1 | 2 | 1 |
| Wingspan (meters) | 1.6 | 2.7 | 1.6 |
| Payload (kilograms) | 0.5 | 1.5 | 1 |
| Maximum takeoff weight (kilograms) | 3 | 17 | 3 |
| engine type | Electric | Fuel | Electric |
| Flight time (minutes) | 90 | 120 | 60 |
| Camera model | ILCE-5100 | ILCE-5100 | ILCE-5100 |
| Number of integrated cameras | 2 | 2 | 1 |
| Focal length (mm) | 30 | 30 | 30 |
| Camera resolution (pixel) | 6000 × 4000 | 6000 × 4000 | 6000 × 4000 |
| Camera model | ILCE-5100 | ILCE-5100 | ILCE-5100 |

*2.3. Environmental Variables*

The WorldClim database encompasses historical, contemporary, and projected future climates, considering various scenarios. This dataset comprises a total of 19 climate factors, yet intercorrelations exist among them, leading to information redundancy and potentially influencing result accuracy. Addressing this, the independent construction of environmental factors allow for the consideration of the diverse impacts on the SDM according to the specificities of the study. Thus, to enhance the calculation precision of the SDM and comprehensively depict the environmental factors influencing the suitable habitats for wildlife from diverse perspectives, this study primarily incorporates the physical geographic factors, climatic factors, food source factors, and human-induced interference factors input parameters into the model, which are presented in Table 2.

The intricate interplay between climate, LUCC, and suitable habitat of a species is complex and influential [38]. Notably, LUCC stands out as a significant determinant affecting the suitable habitat area of species [39]. Moreover, the ECRPSR plays a crucial role in shaping these suitable habitats by restoring LUCC and establishing protected areas. The ECRPSR in YRSNP was launched in 2005 and it entered its second phase of construction in 2015. Accordingly, this study focuses on distinct periods: the pre-project period spanning 2000–2004 (Phase I), the first project phase spanning 2005–2014 (Phase II), and the ongoing second phase covering 2015– 2020 (Phase III). To comprehensively assess the temporal and spatial evolution of suitable habitat areas for large ungulate herbivores in the YRSNP and to determine the impact of the ECRPSR program on these areas, environmental factors are selected based on the different project phases. For the resampling of climatic data, the raster point conversion operation was first carried out for the original 1 km data, and the value of the raster was converted to vector points with an interval of 1 km, and each vector point was regarded as a weather station. The ANUSPLIN meteorological interpolation method was used to interpolate the vector points into meteorological data with a precision of 30 m [40]. This method has been widely used in climatic data processing with good accuracy [41]. In addition, the operation of raster turning point and the method of Kriging interpolation [42] were adopted to convert each raster data into vector points, and obtain the data with 30 m accuracy of human footprint index.

**Table 2.** The environment variables selected by the species distribution modeling (SDM).

| Category | Environment Factors | Abbreviation | Source | Resolution |
|---|---|---|---|---|
| Physical Geographical Factors | Digital Elevation Model | DEM [43] | Geospatial data cloud (https://www.gscloud.cn/, accessed on 23 July 2023) | 30 m |
| | Slope | SLOPE | Extracted from DEM by arcgis 10.8 | 30 m |
| | Land Use and Land Cover Change | LUCC | China Land Cover Dataset [44] | 30 m |

**Table 2.** *Cont.*

| Category | Environment Factors | Abbreviation | Source | Resolution |
|---|---|---|---|---|
| Climatic Factors | Precipitation | PRE | National Earth System Science Data Center (http://www.geodata.cn/, accessed on 23 July 2023) | 1000 m |
| | Temperature | TEM | National Earth System Science Data Center (http://www.geodata.cn/, accessed on 23 July 2023) | 1000 m |
| Food Source Factors | Normalized Differential Vegetation Index | NDVI | National Science & Technology infrastructure (http://www.nesdc.org.cn/, accessed on 23 July 2023) | 30 m |
| | Distance to Water | DW | Extracted from LUCC by arcgis 10.8 | 30 m |
| Human Interference Factors | Human Footprint Index | HFP | A dataset of human footprint over the Qinghai-Tibet Plateau [45] | 1000 m |

### 2.4. Parameters Optimization of MaxEnt Model

Feature combination (FC) and the regularization multiplier (RM) are the two most important parameters of MaxEnt for predicting the potential suitable habitat of species [46]. The optimization of FC and RM do help to significantly improve the accuracy of the model. FC consists of linear (L), quadratic (Q), product (P), threshold (T), and fragment (hinge, H), which can produce 31 different combinations. Generally, the RM parameter is set to less than 4 [47]. The RM value was set at an interval of 0.2 from 0.1 to 4 by using R package of "Kuenm", which can be downloaded at https://github.com/marlonecobos/kuenm, accessed on 27 July 2023. The most statistically significant model with an omission rate of less than 5% was first selected and, according to the Akike information criterion (AICc), the model with the lowest Delta AICc value remained as the best recommended model [48]. The selected models can be seen in Figure 2. As for the Tibetan wild ass, the RM parameter was set to 0.7 and the FC combination was set as Q and P. As for the Tibetan gazelle, the RM parameter was set to 0.5 and the FC combination was set as L and Q. Such parameter settings can effectively avoid overfitting, as well as the mobility from the known distribution region to the predicted region being the best.

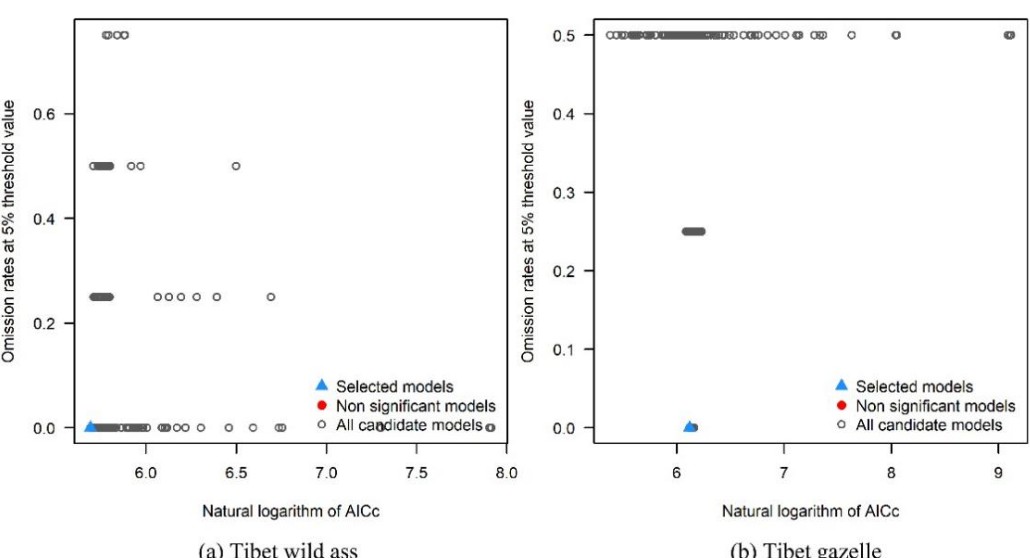

(a) Tibet wild ass    (b) Tibet gazelle

**Figure 2.** Selected models from R program package.

### 2.5. The Evaluation of Suitable Habitat in Different Phases of the ECRPSR

In this study, we incorporated species occurrence sites and environmental factors into the MaxEnt model, with a random inspection percentage set at 25%. In addition, 75% of the species occurrence sites were designated as the training set for model calibration. The

cross-validation repetition number was 10 times and the average serves as the output of the prediction result with the maximum background points number of 10,000.

The accuracy of the model was assessed with the obtained area under the curve (AUC) value. Based on the criteria proposed in Swets' literature, AUC values are within the range of [0.9, 1] which indicate excellent simulation results [49]. To analyze the impact of various environmental factors and determine the main influencing factors on the distribution of species-suitable areas and identify the primary influencing factors, we employed the jackknife method. Additionally, the response curve of each environmental factor was examined to determine its influence range and threshold. By conducting a joint analysis of the jackknife method and the response curve, we were able to assess the contribution of different environmental factors to species distribution and identify the range of influence of dominant environmental factors [43].

The simulation result output is a probability raster layer in ASCII format, representing the potential distribution of the species. The probability values range from 0 to 1, with higher values indicating a greater probability of species distribution. The suitable habitat level is identified with differentiation probability of species distribution. According to similar species research, natural breakpoint method is used to effectively show the level of species distribution probability [26,50]. According to the natural breakpoint method, the suitable habitat can be divided into unsuitable habitat, low suitable habitat, moderately suitable habitat, and highly suitable habitat.

### 2.6. Assessment of Migration Path of Large Wild Herbivores in Different Phases of the ECRPSR

To comprehensively analyze both species' potential habitats and ecological corridors, we employed the MaxEnt-MCR model proposed by Bao, which combines the SDM with migration resistance surfaces [43]. For the selection of ecological source areas, the morphological spatial pattern analysis (MSPA) model was employed, using the highly suitable habitat as the foreground for binarization treatment. The core areas exceeding 5 km$^2$ were then identified as ecological sources. To construct spatial migration paths and corridors for large wild herbivores in different phases, the MCR model [51] was employed, promoting biodiversity and gene exchange among species [52]. The formula for the MCR model is as follows [53]:

$$MCR = f_{min} \sum\nolimits_{j=n}^{i=m} (D_{ij} \times R_i)$$

where MCR represents the minimum cumulative resistance value; $f_{min}$ denotes the positive correlation of ecological process of the minimum cumulative resistance; $D_{ij}$ refers to the spatial distance from ecological source j to i; and $R_i$ indicates the resistance coefficient of landscape unit to biological movement.

This study divided the suitable probability intervals of environmental factors, obtained from the MaxEnt model, into four categories, 10, 20, 30, and 40, representing different resistance values. These values correspond to varying degrees of resistance encountered by the species during migration, with the most suitable interval assigned a value of 10 and the least suitable interval assigned a value of 40. To construct the minimum cumulative resistance surface, we utilized the contribution rate of each environmental factor identified by the MaxEnt model as the weight. Using the cost distance and cost paths module in the ArcGIS 10.8 software, the ecological corridors between any two ecological sources of large wild herbivores were identified.

## 3. Results

### 3.1. The Spatial–Temporal Evolution of Potential Suitable Habitat of Large Wild Herbivores

The AUC values of the MaxEnt model for the Tibetan wild ass and Tibetan gazelle were all greater than 0.9, indicating reliable and valid simulation results across Phases I to III. The contribution rates of environmental factors, depicted in Figure 3, were used as weights to construct the resistance surface for the MaxEnt-MCR model. Notably, precipitation (mean 38.17%), human footprint index (mean 37.2%), and distance to water

(mean 11.13%) emerged as the primary environmental factors influencing the distribution of suitable habitats for the Tibetan wild ass. For the Tibetan gazelle, precipitation (mean 57.33%), air temperature (mean 15.07%), and human footprint index (mean 11.9%) played pivotal roles in determining suitable habitat distribution. Temperature sensitivity was particularly pronounced for the Tibetan gazelle, underscoring the significance of maintaining temperature stability within the YRSNP to protect its suitable habitat. Moreover, human disturbance factors exerted a substantial impact on large wild herbivores in the YRSNP, emphasizing the criticality of mitigating human interference to restore and preserve the quantity and quality of suitable habitat areas for these species.

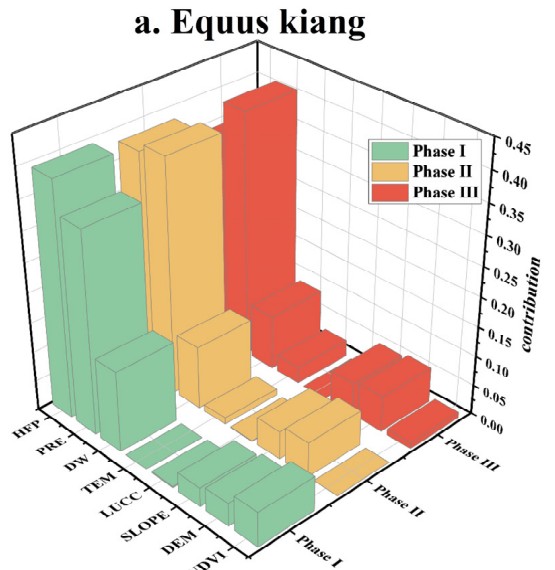 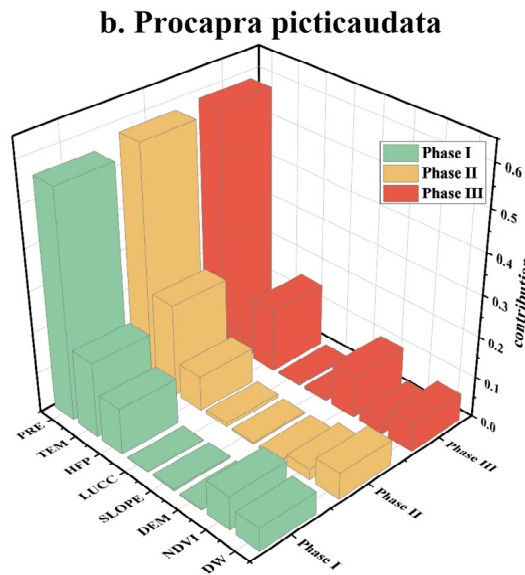

**Figure 3.** The contribution rate of environmental factors in different phases of each species. (**a**) Tibetan wild ass, (**b**) Tibetan gazelle.

The primary habitat of the Tibetan wild ass is predominantly located in the northwestern region of Maduo County, characterized by bands encircling Zhaling Lake, Eling Lake, and Xingxinghai, as shown in Figure 4. Additionally, suitable areas have been identified in the central and northeastern parts of the YRSNP. Notably, areas with high habitat suitability are in close proximity to water sources, which aligns with the MaxEnt model indicating that DW serves as the dominant factor influencing the distribution of the Tibetan wild ass. In the absence of any construction activities, the highly suitable area measured 294.42 km². Following Phase II, the highly suitable area has been effectively restored, encompassing 437.16 km², representing a remarkable increase of 48.5%.

The suitable habitat of the Tibetan gazelle is mainly located in the central and northern regions of Maduo County, exhibiting favorable connectivity. The level of suitability gradually diminishes as the distance from the Zhaling– and Eling Lakes and Xingxinghai areas increases. The habitat covers a total area of 6066.89 km², constituting approximately 31.79% of the total area of the YRSNP. Following the implementation of two phases of the ECRPSR, the habitat area has increased to 6873.33 km², with a noticeable shift of habitat centers toward the south. Notably, the Xingxinghai area serves as a significant habitat for the Tibetan gazelle, encompassing 10.71% of the total area of the YRSNP.

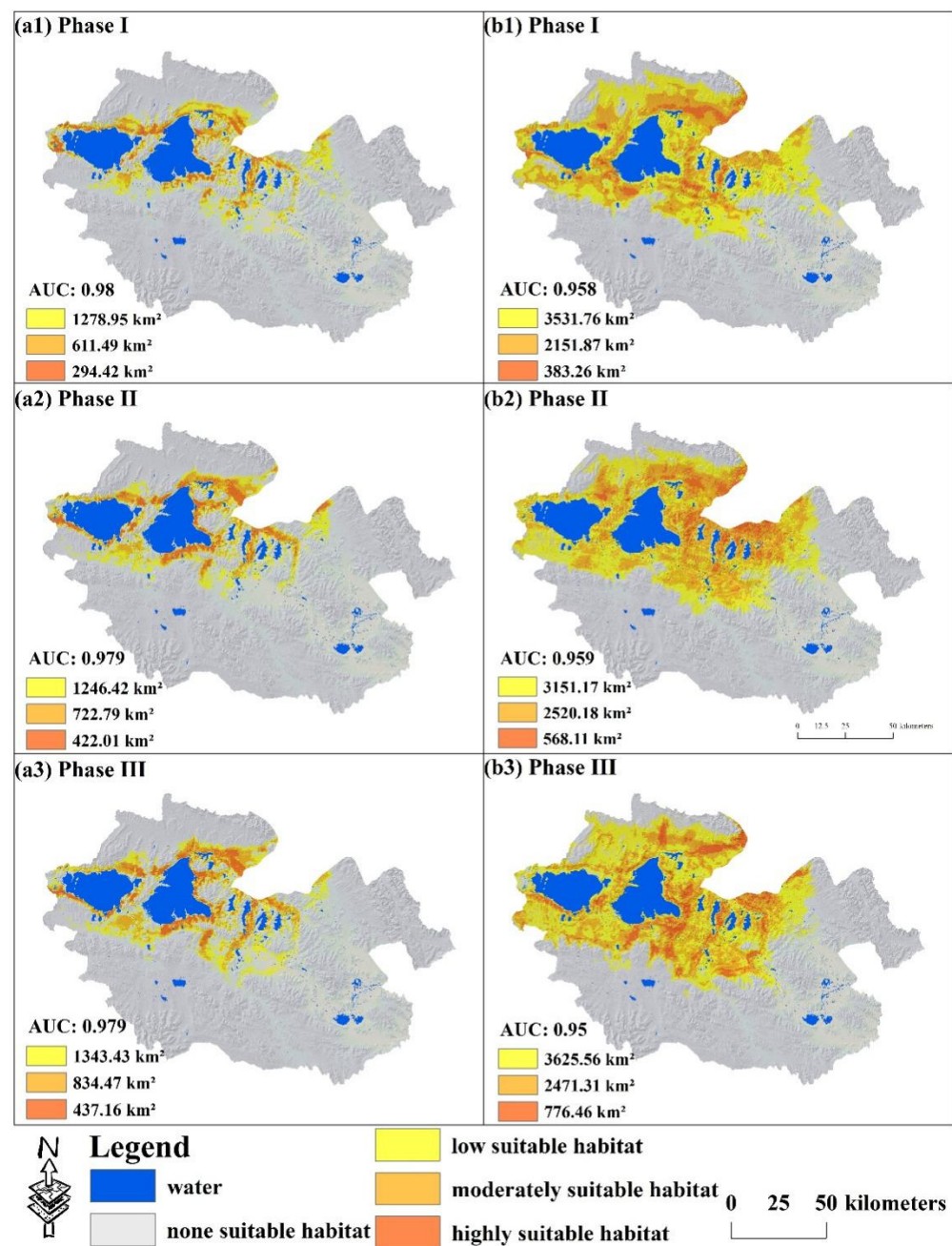

**Figure 4.** The spatio-temporal evolution of species suitable habitat under different phases. (**a1–a3**): Tibetan wild ass; (**b1–b3**): Tibetan gazelle.

### 3.2. The Positive and Negative Change in Species-Suitable Habitat

The changes of area of each level of suitable habitat are calculated and the overlay analysis is used through ArcGIS 10.8 software to make the changes visible, as shown in Figure 5. For the Tibetan wild ass, habitat negative changes were primarily concentrated in the southern region of the Zhaling–Eling Lake area and the northeastern part of the YRSNP during Phases I–II. However, considerable progress was made in Phases II–III, with only a small amount of habitat-negative change observed, primarily in the abdominal area of Maduo County. As for the Tibetan gazelle, habitat-positive change during Phases I–II was mainly concentrated in the southeast of the Xingxinghai area, while further positive change and protection efforts were carried out in Phases II–III. Moreover, a notable increase in suitability was observed in the northern part of the Zhaling–Eling Lake area. Overall, the

implementation of the ECRPSR led to a considerable augmentation in the distribution of large wild herbivores in the YRSNP during Phases II–III compared to Phases I–II.

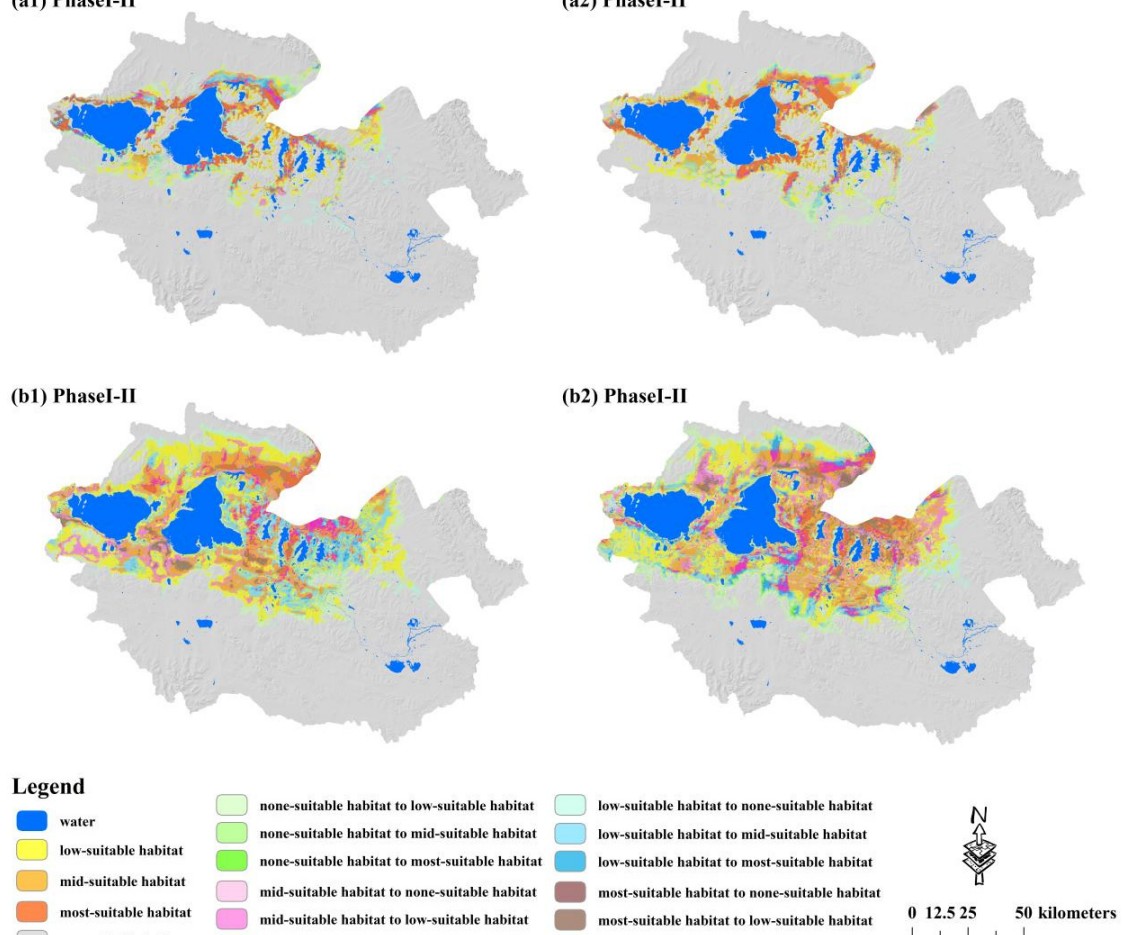

**Figure 5.** The positive and negative change in species suitable habitat under Phase I–II and Phase II–III. (**a1**,**a2**): Tibetan wild ass; (**b1**,**b2**): Tibetan gazelle.

In general, the invariant area of habitat suitability for the Tibetan wild ass surpasses that of the Tibetan gazelle. Nevertheless, the net growth area (positive changes subtract negative changes) for the Tibetan wild ass constitutes 15.95% (Phases I–II) and 13.13% (Phases II–III) of the total habitat suitability area. In comparison, the net growth area for the Tibetan gazelle accounts for 31.18% (Phases I–II) and 28.10% (Phases II–III) of the total habitat suitability area. During Phase I–II, the invariant area of the Tibetan wild ass and Tibetan gazelle are 1126.91 km$^2$ ($p < 0.01$) and 3372.61 km$^2$ ($p < 0.01$), respectively. The net growth areas of the Tibetan wild ass and Tibetan gazelle are 434.05 km$^2$ ($p < 0.05$) and 848.34 km$^2$ ($p < 0.05$). The negative change in habitat for the Tibetan gazelle exhibits greater severity. Specifically, the negative change area in the YRSNP comprises 6.47% during Phases I–II and 8.88% during Phases II–III.

### 3.3. The Construction of Ecological Corridors in Different Phases of the ECRPSR

The contribution rates of environmental factors identified by Tibetan wild ass and Tibetan gazelle are shown in Figure 6. The environmental contribution of DEM, SLOPE, DW to the Tibetan wild ass decreased with the increase in the range, indicating that an altitude that is too high, slope, and distance from the water are not conducive to a suitable habitat for the Tibetan wild ass. At an altitude of about 4500 m, a slope of 32.5 degrees, and being 17,000 m away from the water, the development of the Tibetan wild donkey will be

very unfavorable. In addition, the suitable temperature and precipitation will be conducive to the development of the Tibetan wild ass, and the suitable range is about −4~−2 °C (TEM) and 2500~5000 mm (PRE). The suitable habitat area of the Tibetan gazelle is less affected by slope change, and any slope is more suitable for Tibetan gazelle. In addition, the increase of temperature will help to increase the probability of suitable habitat, and the suitable range is −3 °C~4 °C. The appropriate distance from the water and the appropriate precipitation are the key factors affecting the suitable habitat area of the Tibetan gazelle, and the suitable range is 2500 m~12,500 m (DW) and 2500 mm~4500 mm (PRE).

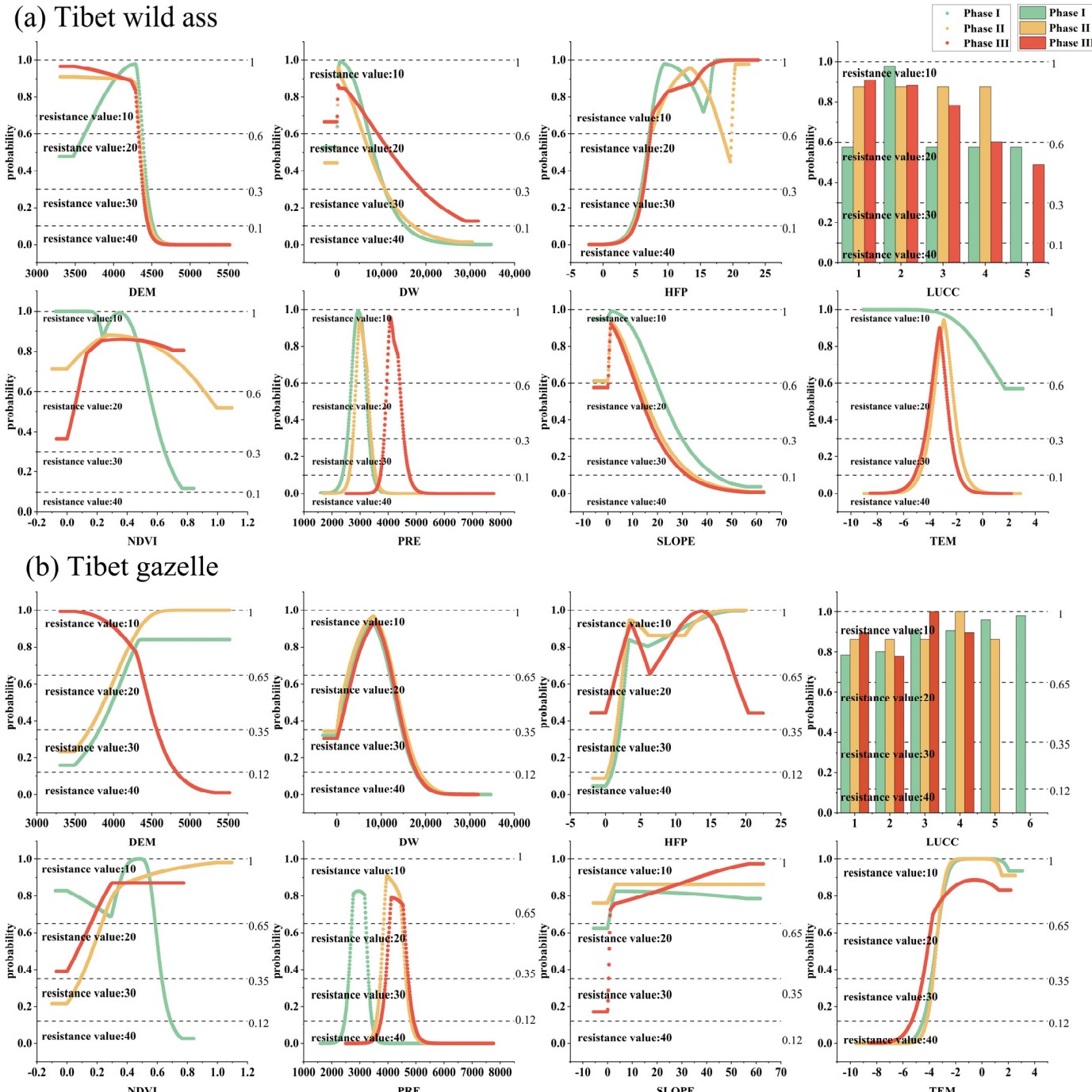

**Figure 6.** The contribution threshold of environmental factors with differentiation resistance value at different phases. (**a**) Tibetan wild ass, (**b**) Tibetan gazelle.

In the absence of the ECRPSR, the ecological source areas for the Tibetan wild ass and Tibetan gazelle predominantly centered around Zhaling and Eling Lakes, forming a circular ecological corridor encompassing the two lakes, as shown in Figure 7. Notably, the Zhaling–Eling Lake region emerged as a vital ecological concentration area warranting attention. During the first phase of the ECRPSR, the southern portion of the lake exhibited a significant increase in the ecological source area for the Tibetan wild ass, while the ecological source area for the Tibetan gazelle predominantly expanded in the Xingxinghai area. Therefore, the central axis of the ecological migration corridor shifted southward. In the subsequent phase, substantial patches of ecological source areas for the Tibetan wild ass were observed in the southern region of Eling Lake, with a primary concentration in the southern part of the Xingxinghai area. The geographical expansion of species is predominantly occurring toward southern areas, leading to an increase in the diversity of species corridors. However, the ecological source located in the southwestern region of the Zhaling Lake remains secluded, exhibiting limited connectivity with other ecological sources.

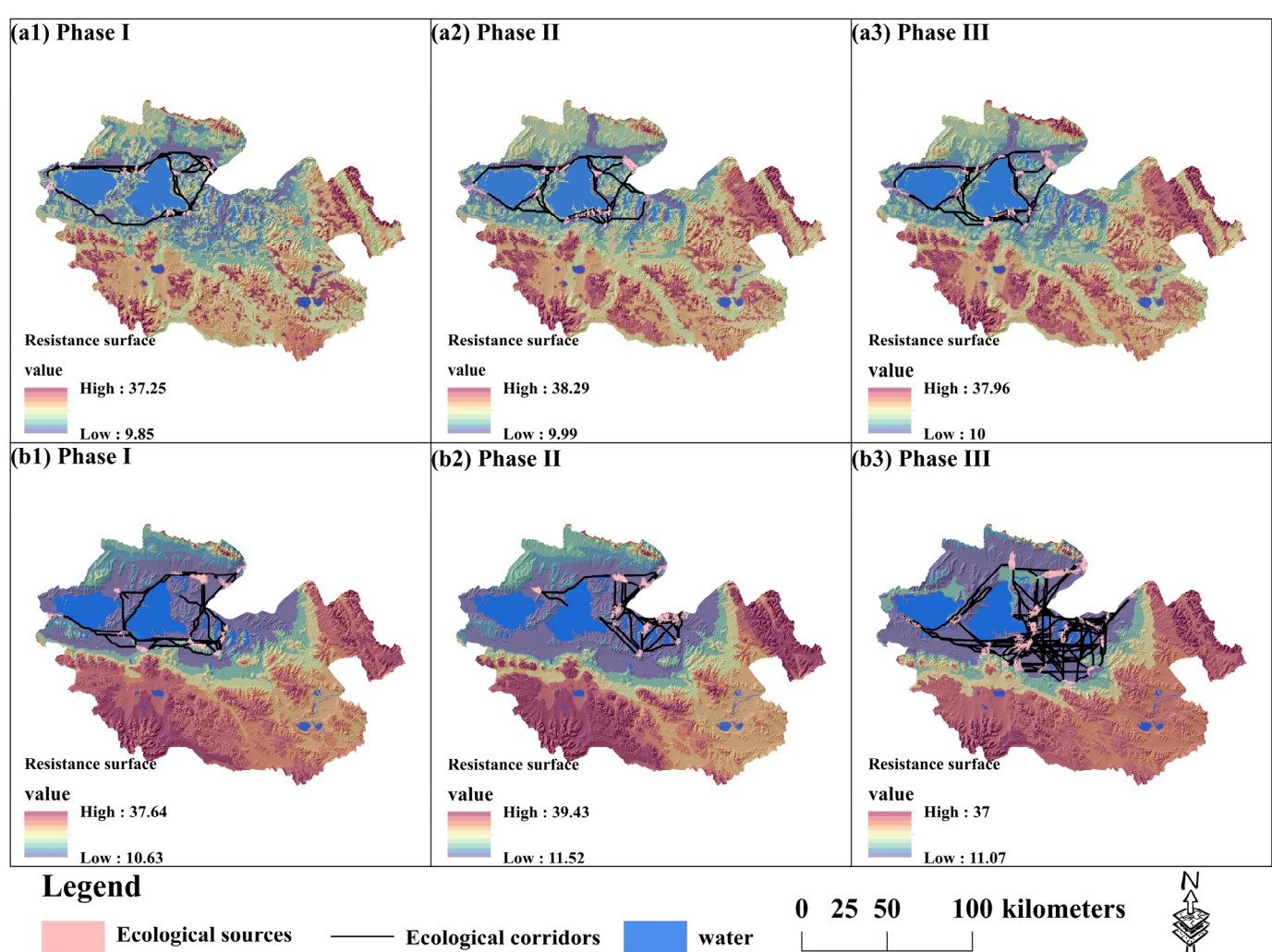

**Figure 7.** The ecological corridors of large wild herbivore at different phases. (**a1**–**a3**): Tibetan wild ass, (**b1**–**b3**): Tibetan gazelle.

## 4. Discussion

### 4.1. Comparison of Species-suitable Habitat with Previous Studies

The findings of suitable habitat maps from Li et al. [54] and Gao et al. [55] align closely with our own research. However, it should be noted that while these studies encompass the YRSNP, they lack specific simulations. These studies suggest that the southeastern portion of the YRSNP also serves as a suitable habitat for large wild herbivores, which show

differences with our findings. The underlying causes for this phenomenon can be attributed to several factors. (1) The selection of driving factors plays a crucial role in the species distribution modeling process [56]. Many studies have utilized the Worldclim database for driving factor selection and conducted correlation analyses to mitigate variable autocorrelation [26,35]. However, these selected variables possess limitations in accurately reflecting climate change. In our research, a multi-dimensional approach was adopted, incorporating climate, environment, food availability, human disturbances, and other variables to conduct the SDM. It is worth noting that in the SDM process even within the same region and for the same target species, variations in environmental parameters yield different results, which be exemplified by the research findings of Zhang and Shi [26,57]. (2) The spatial scale of simulation plays a significant role in contributing to deviations in SDM results. Existing evidence has demonstrated that different scales of study areas leads to contrasting results, as highlighted in Zhang's research on Tibetan gazelles [55,57,58]. Moreover, these studies primarily employed 1 km data to model species distribution in similar regions [54], yet yielded dissimilar results. This can be attributed to the relatively macroscopic nature of the study areas, lacking specific simulations of the YRSNP, which further contributes to result discrepancies. Previous literature has extensively examined the influence of spatial data resolution on SDM, and the findings provide robust evidence supporting the impact of spatial resolution on SDM results [59,60]. (3) The species occurrence sites significantly affects the accuracy of SDM. While the accuracy of the model remains high in small samples, it is crucial to recognize that the species distribution points significantly influence the identification of potential habitats. Currently, studies on species occurrence sites primarily rely on databases [26], line transects [38], GPS [30], and UAVs [28,43]. However, it should be noted that the species occurrence sites obtained from databases tend to be macroscopic in nature, leading to distortions in regional species distribution patterns.

*4.2. Adjustment of Protected Area Boundaries by Setting a Wildlife Refuge*

Certain nature reserves exhibit deficiencies in terms of their scope demarcation and functional zoning, lacking scientific and rational approaches. The omission of crucial habitats for wildlife within these reserves remains a concern [23], with persistent conflicts arising between wildlife and livestock in the core areas [38]. Moreover, the efficacy of targeted and operational control measures are not sufficiently robust. To address wildlife protection concerns comprehensively, there is an urgent need to optimize and readjust the scope and functional zoning of nature reserves, while concurrently integrating and streamlining diverse protected areas.

The protected areas within the ECRPSR primarily encompass the Zhaling–Eling Lake Natural Reserve and Xingxinghai Natural Reserve (Figure 8). These reserves have been designated with the objective of rejuvenating ecosystems, with the aim of mitigating or reversing grassland and wetland ecosystem degradation [61]. After analyzing the spatial distribution, it is evident that the prime habitat for Tibetan wild asses is concentrated predominantly within the core area of the Zhaling–Eling Lake Natural Reserve, although there are smaller distributions in other regions. On the other hand, the suitable habitat for Tibetan gazelles is primarily concentrated within the core and buffer areas of the Zhaling–Eling Lake Natural Reserve, as well as the western region of the Xingxinghai Natural Reserve. The precise impact of climate change on species abundance remains uncertain [4,62]. Nevertheless, species abundance distribution can serve as a valuable tool for assessing species coexistence patterns and guiding targeted conservation efforts aimed at enhancing species diversity. Notably, a predominant concentration of species richness is observed in the vicinity of the Zhaling–Eling Lake area and the Xingxinghai area, covering an expansive area of 2536.33 km$^2$. It is noteworthy that the distribution area of a single species accounts for 23.23% of the total suitable area, primarily situated at the boundaries of suitable habitats. Conversely, in the southern portion of the test area within the Zhaling-Eling Lake Natural Reserve and the eastern region of the Xingxinghai Natural Reserve, no suitable areas were identified for Tibetan wild asses and Tibetan gazelles.

Therefore, future ECRPSR efforts within the YRSNP should prioritize these specific areas and expand suitable herbivore habitats accordingly.

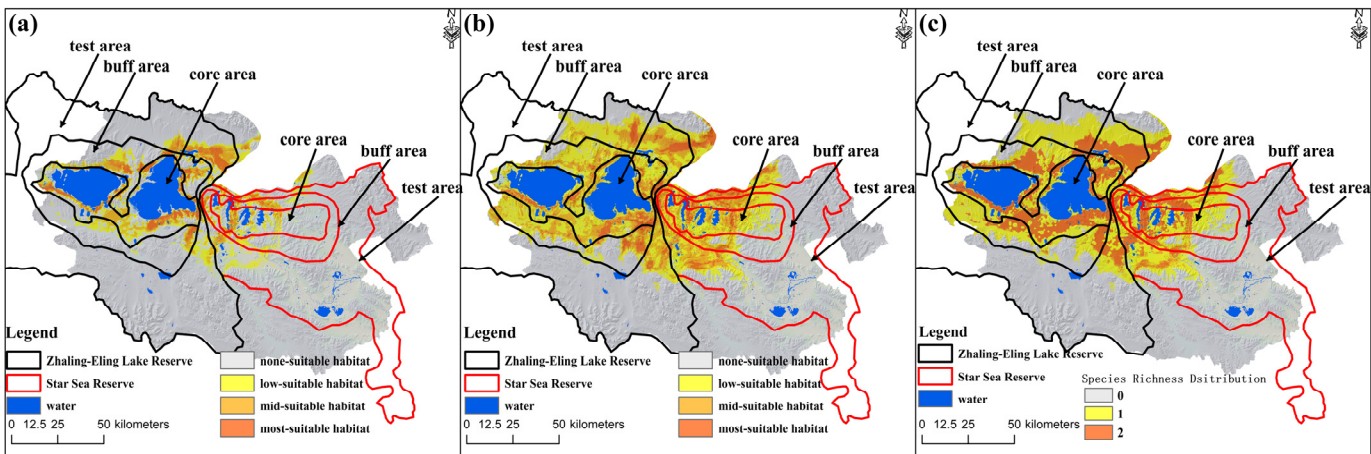

**Figure 8.** The natural reserve of the YRSNP. (**a**) Suitable habitat of Tibetan wild Ass, (**b**) Suitable habitat of Tibetan gazelle, (**c**) Species richness distribution.

Functioning as interconnected entities, the core conservation and the natural reserve based on the SNP plan play vital roles in spatial planning, as demonstrated in Figure 8. Within the boundaries of the reserve, the core conservation area spans 3258.70 km$^2$, accompanied by a buffer zone measuring 3467.94 km$^2$ and a designated test area spanning 1297.58 km$^2$. Notably, the species-suitable habitat extends to the northern section of the Zhaling–Eling Lake and the southern region of the Xingxinghai area, despite their exclusion from the core conservation. Instead, these areas fall within the buffer zone of the Zhaling–Eling Lake Natural Reserve. Hence, it becomes necessary to revise the boundary of the core conservation in the YRSNP and establish well-defined rights and responsibilities regarding wildlife protection through targeted control measures.

The species-suitable habitat area has a certain functional relationship with its population size. The research shows that a wide suitable habitat area will promote species reproduction and communication, thus promote the growth of its population size [63]. However, with the increase in the most representative large wild herbivores such as the Tibetan wild ass and the Tibetan gazelle, the conflict between grass and livestock in the YRSNP will become more and more prominent [55]. The competitive eating conflict between human and wild animals will either cause a disadvantage in the biodiversity conservation or a downward economic development problem [64]. In addition, a large number of herbivores will also lead to the consumption of grassland, shrubs and other land resources, which destroy the landscape of the YRSNP. Especially in the marginal areas outside the protected boundaries, resources will be further consumed with the contradictions between grass and livestock and between people and wild animals further prominent [38]. Therefore, adjusting the core conservation boundary will directly help to protect the species' population and avoid further intensification of the conflicts between wildlife and domestic animals, which will help separate human and wildlife behavior boundaries while promoting biodiversity and economic development, the adjustment of the core conservation boundary is shown in Figure 9.

In recent years, global warming has necessitated a re-evaluation of protected area establishment, as it has the potential to induce rapid shifts in species distribution and could eventually serve as the sole refuge for critically endangered species in the future [65]. Adapting protected area boundaries based on species' ecological corridors, which are indicative of their connectivity, represents an effective strategy for conserving species diversity [66]. To provide further protection for the species within the YRSNP, the establishment of a wildlife reserve is proposed, as depicted in Figure 9. This reserve would predominantly

encompass the northern section of the Zhaling–Eling Lake Natural Reserve's test area and the southern region of the Xingxinghai Natural Reserve's test area, building upon the foundation of the core conservation. By designating an exclusive species reserve in the YRSNP, we can effectively address conflicts arising from functional zoning plans, integrate various protected areas, clarify clear rights and responsibilities pertaining to wildlife protection, and protect wildlife habitat distribution.

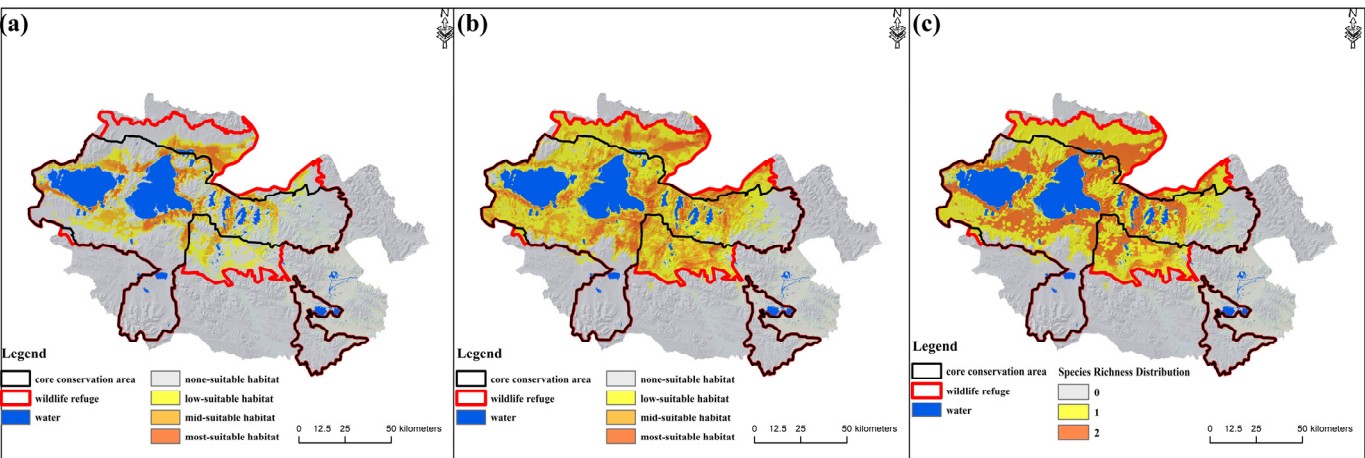

**Figure 9.** Core conservation and boundary adjustment. (**a**) Suitable habitat of Tibetan wild ass, (**b**) suitable habitat of Tibetan gazelle, and (**c**) species richness distribution.

### 4.3. Suggestions for Protecting Biodiversity in the YRSNP

The future distribution of wildlife will be greatly influenced by their capacity to adapt or tolerate diverse climatic conditions. Species that exhibit heightened sensitivity to climatic factors, such as drought, heavy rainfall, and temperature fluctuations, are particularly vulnerable and face the risk of habitat loss and extinction [67]. Extensive research has demonstrated that human-induced disturbances constitute a primary driver of species decline and potential extinctions [64]. Our results highlighted the significant roles of PRE and TEM in shaping the habitat distribution of large wild herbivores. Additionally, the HFP emerged as a crucial environmental parameter elucidating changes in suitable habitat for large wild herbivores in the YRSNP, accounting for 37.2% (Tibetan wild ass) and 11.9% (Tibetan gazelle). Therefore, for species exhibiting sensitivity to environmental dynamics, stabilizing climate change and minimizing human-induced disruptions are pivotal for the preservation of species diversity in the YRSNP.

The capacity of species to access other suitable habitat areas predominantly relies on the connectivity between ecological sources and the species' migratory abilities [68]. However, enhancing the adaptive capacities of species to cope with climate change and migration challenges cannot be achieved within a short timeframe; rather, it necessitates multiple generations of natural selection and adaptation processes [39]. Therefore, governmental intervention through the establishment of an ecological compensation mechanism and the creation of protected areas emerges as a crucial approach to mitigating climate change effects and sustaining ecological equilibrium [24]. Specifically, these animals have displayed enhanced resistance in the southeast direction and improved connectivity toward the northwest. This pattern can be attributed to the fragmentation of the ecological source areas for the Tibetan wild ass and Tibetan gazelle, as well as the decline in habitat suitability at the park boundaries. While the ECRPSR has proven effective in ecosystem restoration, it is necessary to concurrently focus on constructing ecological corridors that align with the trajectory of climate change to counteract the adverse impacts caused by resistance growth.

The findings indicate the susceptibility of the ecosystem at the species' habitat boundary to climate change, rendering it highly responsive. While numerous studies have examined the species-suitable habitat changes under various scenarios, research pertaining to the

species-suitable habitat boundaries remains insufficient. The loss of habitat in these boundary areas directly contributes to population decline, consequently disrupting the dynamics of predator–prey interactions [69]. Moreover, the effectiveness of establishing protected areas is contingent upon the habitat boundary's integrity. It is necessary to impose limitations on the proportional harvest rate at the boundary of a protected area to guarantee the genuine long-term sustainability of species within its confines [70]. Consequently, directing attention toward vulnerable habitat boundaries and implementing essential conservation measures will contribute to the preservation of sustainable development for species.

*4.4. Innovations, Limitations, and Prospects*

In this research, UAVs were utilized to examine the spatial distribution of large wild herbivores within the YRSNP, while the MaxEnt model was employed to analyze the habitat distribution and principal influencing factors for the Tibetan wild ass and Tibetan gazelle, disregarding the impact of the ECRPSR and its first and second phases. Additionally, the MaxEnt-MCR model was employed to assess species-suitable habitat area changes across the phases and identify changes in ecological corridors for large wild herbivores. However, it is important to note that certain factors such as poaching [71], fencing [72], and extreme weather events [73] were not accounted for in the SDM process, potentially yielding detrimental effects on species populations. Future studies should seek to simulate the suitable habitat distribution of large wild herbivores under diverse climatic conditions and establish scenario-based analyses and comparisons, thereby furnishing scientific insights for species diversity conservation and ecological restoration projects. In addition, the interpolation method applied in the research may lead to data distortion, regional inaccuracy, and other problems. Therefore, the next step of the research will also focus on producing high-precision regional climate and environmental data and human activity data, which is beneficial to establishing an integrated monitoring platform in Yellow-River-Source National Park.

**5. Conclusions**

Located in the hinterland of the Sanjiangyuan region, the YRSNP is situated in one of the ecologically vulnerable zones, characterized by climate change-induced challenges, particularly with regard to the ecological environment and species diversity. The findings of this study clearly indicate that the impact of the ECRPSR has effectively protected populations of large wild herbivores and their suitable habitats within the YRSNP. Therefore, we can draw conclusions as follows:

1. In light of the influence of the ECRPSR and SNP, the ecological corridors primarily concentrate within the core areas of the Zhaling–Eling Lake Reserve and the Xingxinghai Reserve, whereas the buffer area and test area lack such corridors. It is necessary to undertake ecological restoration efforts within these buffer and test areas to ensure the preservation of habitat distribution for large wild herbivores.
2. The clarification of wildlife protection rights and responsibilities, as well as the implementation of targeted control measures for biodiversity conservation and focus on the marginal areas of species-suitable habitats which are particularly susceptible and prone to loss, can be effectively achieved through the adjustment of wildlife natural reserve boundaries based on the core conservation area of the YRSNP. The intervention of the ECRPSR becomes necessary to protect ecological sources.
3. Given the significant impacts of climate change, which have led to alarming declines and extinctions of species, it is crucial to stabilize the changing climate and establish ecological corridors that align with its trajectory.

**Author Contributions:** Conceptualization, F.Y.; methodology, S.B.; validation, F.Y. and S.B.; formal analysis, S.B.; writing—original draft preparation, S.B.; writing—review and editing, S.B.; supervision, F.Y.; funding acquisition, F.Y. All authors have read and agreed to the published version of the manuscript.

**Funding:** This research was funded by National Natural Science Foundation of China, 42001235, F.Y.

**Data Availability Statement:** The raw data supporting the conclusions of this article will be made available by the authors on request.

**Acknowledgments:** We would like to express our sincere gratitude to the editors and two anonymous reviewers for their valuable comments, which have greatly improved this paper. We also would like to acknowledge Quanqin Shao of the Institute of Geographic Sciences and Natural Resources Research, CAS for providing the UAV survey data.

**Conflicts of Interest:** The authors declare no conflicts of interest.

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
