# Peer review of "Identification of Potential Habitats and Adjustment of Protected Area Boundaries for Large Wild Herbivores in the Yellow-River-Source National Park, China"

_land, doi:10.3390/land13020186_

Round 1

Reviewer 1 Report

Comments and Suggestions for Authors

Dear authors,

I have reviewed your paper entitled ‘Identification of Potential Habitat and Adjustment of Protected Area Boundaries for Large Wild Herbivore in the Yellow River Source National Park, China’. The paper focuses on assessing the impact of climate change and human activities on the habitat of Tibetan wild ass and Tibetan gazelle in the Yellow River Source National Park (YRSNP). It uses unmanned aerial vehicles (UAV) for data collection and employs the MaxEnt model for habitat assessment. Overall, the use of UAVs for data collection and the MaxEnt model for analyzing habitat suitability are innovative and provide a detailed understanding of species distribution. However, I have several concerns about the methodology of this work.

Firstly, resampling climate and human interference factors from a 1000m resolution to 30m using simple interpolation is not the right way. This resampling practice can lead to significant overestimation of spatial precision. Such a coarse resolution data may not accurately reflect the fine-scale variability and complexity of environmental factors at a smaller scale. I recommend that the authors provide a thorough justification for this resampling approach, including its scientific rationale and potential impacts on the outcomes.

Second, more details are needed to strengthen the credibility of the MaxEnt model, such as the number of background points, the number of repetitions of cross-validation, the class of features (can be tunned by some R package), regularization parameters (can be tunned by some R package), etc.

Elaborate on how the findings can directly influence conservation policies and practices in the Yellow River Source National Park. This is because national parks do not target just two species of wildlife, but also other biological groups, landscapes, and resources.

Best wishes

Comments on the Quality of English Language

The overall English of this article is quite fluent

Author Response

Dear reviewer:
Thank you for your letter and for the reviewers’ comments concerning our manuscript entitled “Identification of potential habitat and adjustment of protected area boundaries for large wild herbivore in the Yellow River Source National Park, China” (ID: Land-2846977). Those comments are all valuable and very helpful for revising and improving our paper, as well as the important guiding significance to our study. We have studied comments carefully and have made correction which we hope meet with approval. Revised portion are marked in red in the paper. Please see the attach file.

Reviewer 2 Report

Comments and Suggestions for Authors

A fairly solid study regarding the habitat availability and spatial distribution of two selected large herbivorous mammal species in and around a protected area in central China. Definitely worth being published in the journal Land given some minor adjustments and corrections. To me – a rather technical paper with surprisingly little reference to the species studied – instead, the focus is very much on the method – I am missing some statements about the biology of the two investigated species, e.g. in terms of the ecological plasticity and the differences in feeding ecology also owing to different phylogenetic and ecological adaptations…etc…

Title: The title is not ideal, as it insinuated that the study is based on several species of large herbivores. In fact, the study deals with exactly two species: Procapra picticaudata and Equus kiang. I thus suggest adjusting the title accordingly, e.g.; “two selected large herbivore species” instead of “large wild herbivore”…

Abstract, L. 15-16: Please present the scientific name when first mention a species name – even in the abstract.

Introduction: Lines 75-76: A general rule concerning (not only) zoological nomenclature: scientific names are always written in italics! Please adjust accordingly.

Line 149: The caption of this figure is not sufficient – the map shows more than just the study area - please add some relevant info, e.g. make reference to the colors and digits inserted…

Lines 259 – 261: Authors write: “The primary habitat of the Tibetan wild ass is predominantly located in the north-western region of Maduo County, characterized by bands encircling Zhaling Lake, Eling Lake, and Xingxinghai, as shown in the Figure 3.” Since Figure 3 does not make reference to the names of the lakes, this can be hardly judged by the reader – suggest to label those lakes (and/or areas) in Figure 1.

Line 268-269: Suggest rewording to avaid redundant use of “primar(il)y”.

Figure 3: I don’t think that each section / map of this figure needs an own north arrow and scale bar – one north arrow and scale bar would be enough. The same applies to the legend of colors – please avoid redundancy. The size of labelling letters could be increased for better readability.

Figure 5: Needs adjustments – the labelling of axes can hardly be read – the letters are way too small – it is not really clear what “a” and “b” stand for. Also, it is not clear why the figure for “PRE” and “SLOPE” on the bottom differ from all others in terms of y-axes-length and auxiliary lines…

Figure 6: Please refer to my comments concerning Figure 3 – the same applies here – kindly adjust.

Line 347: I assume that you mean “discussion” but not “results”?

Line 349: “maps from Li and Gao” should be “Li et al. and Gao et al.” because more than just one author were involved in the maps referred to…

Please see my earlier comment of italic spelling of scientific names – that also applies to the reference section – please revise carefully, e.g., Line 579…

Comments on the Quality of English Language

I am not a native speaker and consider the English appropriate for a scientific text.

Author Response

(The authors gave the same response as above.)
